# Small-Molecule Chemical Knockdown of MuRF1 in Melanoma Bearing Mice Attenuates Tumor Cachexia Associated Myopathy

**DOI:** 10.3390/cells9102272

**Published:** 2020-10-11

**Authors:** Volker Adams, Victoria Gußen, Sergey Zozulya, André Cruz, Anselmo Moriscot, Axel Linke, Siegfried Labeit

**Affiliations:** 1Laboratory of Molecular and Experimental Cardiology, TU Dresden, Heart Center Dresden, 1307 Dresden, Germany; victoria.gussen@gmx.de (V.G.); Axel.linke@tu-dresden.de (A.L.); 2Dresden Cardiovascular Research Institute and Core Laboratories GmbH, 01067 Dresden, Germany; 3Department of Drug Research, Enamine-Bienta Ltd., 02000 Kiev, Ukraine; s.zozulya@enamine.net; 4Department of Anatomy, Institute of Biomedical Sciences, University of Sao Paulo, Sao Paulo 05508-000, Brazil; andrecruz@usp.br (A.C.); moriscot@usp.br (A.M.); 5Medical Faculty Mannheim, University of Heidelberg, 68167 Mannheim, Germany; labeit@medma.de

**Keywords:** cancer cachexia, melanoma tumors, muscle wasting, mitochondrial metabolism, MuRF1, chemical biology

## Abstract

Patients with malignant tumors frequently suffer during disease progression from a syndrome referred to as cancer cachexia (CaCax): CaCax includes skeletal muscle atrophy and weakness, loss of bodyweight, and fat tissues. Currently, there are no FDA (Food and Drug Administration) approved treatments available for CaCax. Here, we studied skeletal muscle atrophy and dysfunction in a murine CaCax model by injecting B16F10 melanoma cells into mouse thighs and followed mice during melanoma outgrowth. Skeletal muscles developed progressive weakness as detected by wire hang tests (WHTs) during days 13–23. Individual muscles analyzed at day 24 had atrophy, mitochondrial dysfunction, augmented metabolic reactive oxygen species (ROS) stress, and a catabolically activated ubiquitin proteasome system (UPS), including upregulated MuRF1. Accordingly, we tested as an experimental intervention of recently identified small molecules, Myomed-205 and -946, that inhibit MuRF1 activity and MuRF1/MuRF2 expression. Results indicate that MuRF1 inhibitor fed attenuated induction of MuRF1 in tumor stressed muscles. In addition, the compounds augmented muscle performance in WHTs and attenuated muscle weight loss. Myomed-205 and -946 also rescued citrate synthase and complex-1 activities in tumor-stressed muscles, possibly suggesting that mitochondrial-metabolic and muscle wasting effects in this CaCax model are mechanistically connected. Inhibition of MuRF1 during tumor cachexia may represent a suitable strategy to attenuate skeletal muscle atrophy and dysfunction.

## 1. Introduction

The term cancer cachexia (CaCax) was introduced to describe a heterogeneous group of syndromes where malignancy is the underlying cause of a general cachexia. CaCax in patients is highly variable and depends on the underlying tumor as well as its treatment [1,2,3,4]. Solid tumors of the pancreas, lung, liver, and gastrointestinal tract are typically associated with cancer cachexia [3]. Overall, it has been estimated that about 50–80% of all malignancies include cancer cachexia, and cancer cachexia may account for as much as 20% of all cancer deaths [2]. Cachexia progression has also been correlated with deteriorated response to chemotherapy [2], and conversely, chemotherapy tends to exacerbate myopathy and cachexia further, often forcing the discontinuance of treatment [5,6,7]. 

Currently, there are no Food and Drug Administration (FDA) approved treatments available for cancer cachexia. In particular, it is associated with myopathy, and often further exacerbated during treatment, which is an urgent unmet medical need. While megestrol, an anabolic steroid, has been approved for cachexia treatment in principle, approval so far has been limited to cachexia caused by Acquired Immunodeficiency Syndrome (AIDS). A recent report on its use in palliative cancer care concluded that megestrol increased appetite, but not the quality of life [8]. Furthermore, its use was associated with thromboembolic events and adrenal suppression [8]. Ghrelin agonists are currently being investigated for their potential benefits in cancer cachexia, based on the rationale that they stimulate appetite. In the ongoing ANTHEM (Autonomic Neural regulation Therapy to Enhance Myocardial function) study, a phase 3 clinical trial, the ghrelin agonist anamorelin is being tested in patients with lung mesothelioma, as this malignancy causes severe cachexia [9].

Loss of appetite and insufficient dietary uptake of amino acids may also contribute to myopathy in malignancies because amino acid deprivation promotes muscle catabolism by activating the atrophic pathways, including the ubiquitin E3 ligases MaFBx and MuRF1, respectively [10,11,12]. Therefore, dietary amino acid supplementation, in particular including the branched-chain amino acids (BCAAs) valine, leucine, and isoleucine, may improve muscle atrophy in cancer cachexia, similar to their well-documented benefits in attenuating sarcopenia during healthy aging [13]. The underlying mechanisms of how BCAAs slow down sarcopenia include downregulation of the muscle-specific E3 ligase MuRF1 [14,15], and this, in turn, attenuates the loss of contractile proteins [16,17,18,19].

Melanoma is the deadliest form of skin cancer, and its five-year survival drops to 5–19% when metastases are present [20]. Furthermore, the development of melanomas is associated with muscle wasting and strength reduction, which could recently be recapitulated in an animal model using B16F10 melanoma cells injected into C57BL/6N mice [21]. Therefore, the B16F10 melanoma mouse model may be a useful model to test new pharmacological treatments in cancer targeting muscle wasting.

We hypothesized that a recently discovered small molecule (ID#704946) from a titin-MuRF1 interaction high-throughput screen, attenuating MuRF1 and MuRF2 protein expression and skeletal muscle atrophy (prominently seen in the tibialis anterior muscle) and muscle dysfunction of the diaphragm muscle in murine cardiac cachexia models [22,23] might also be beneficial to skeletal muscle in CaCax: Skeletal muscles during cancer cachexia also upregulate the MuRF1-UPS (ubiquitin proteasome system) catabolic system [1,2,3,4,5]. For testing MuRF1 inhibitors in a cancer cachexia setting, we injected B16F10 melanoma cells into mouse thighs and monitored melanoma tumor outgrowth until severe cachexia was evident (“Tu mice”). In the present study, two distinct inhibitors (MyoMed-946 and MyoMed-205) were fed to Tu mice, and muscle functions were compared between the different groups until sacrifice. MyoMed-946, although structurally identical to ID#704946 used in former studies [22,23], was obtained by a different large scale synthesis method developed by Enamine–Kiev. MyoMed-205 was obtained by a chemical modification where an ester bond in ID#704946 was replaced by an amide bond, in an attempt to enhance resistance to serum esterase. 

## 2. Methods and Materials

### 2.1. Cell Culture of B16F10 Melanoma Cells and Quantification of Tumor Size during Outgrowth

B16F10 melanoma cells were cultured in Roswell Park Memorial Institute (RPMI)-1640 medium supplemented with 10% fetal calf serum (FCS) at 37 °C and 5% CO_2_. As soon as the cells reached confluency, cells were harvested by trypsin-digestion (0.05% trypsin- Ethylenediamine tetraacetic acid (EDTA) solution; stock purchased from PAA Laboratories GmbH, catalog #L11-003). After centrifugation (10 min at 300× *g*), the cell pellet was resuspended in RPMI-1640 medium without FCS and adjusted to a stock solution containing 5 × 10^6^ cells /mL. After injection into mice (see next section), tumor outgrowth was monitored by caliper: The diameters of the tumors were measured using a caliper on days 9, 10, 12, 15, 17, 19, and 22 after cell inoculation. The volume was calculated using the ellipsoid equation: V = (W × H × L) × 0.5 (V = tumor volume, W = tumor width, H = tumor height, L = tumor length all in mm).

### 2.2. Animal Study Design

Forty C57BL/6N male mice were included in the present study (Figure 1A). At the age of 8 months (bodyweight 25.1 ± 1.1 g), tumor growth was induced in 30 mice by injecting 5 × 10^5^ melanoma cells/per animal, suspended in 100 µL RPMI-media without fetal calf serum (FCS) and injected subcutaneously into their right thighs. As controls, ten animals received 100 µL RPMI-media injections without B16F10 cells. Three days after tumor cell inoculation, the animals were randomized into 3 different groups: (1) animals receiving no treatment and fed with the standard rodent chow (tumor group (Tu), *n* = 10) obtained from Sniff (Sniff GmbH, Soest, Germany, Cat No: 1534-70); (2) ten animals were changed onto chow supplemented with the compound Myomed#946 (0.1% addition to chow; Tu-946); (3) ten animals received chow supplemented with the MuRF1 inhibitor Myomed#205 (0.1% compound supplement to normal mouse chow (Tu-205), which is roughly 25-fold below LD_50_). At days 9, 16, and 23, muscle function was assessed by wire hang tests (WHTs): A standard wire hang construction was used with a wire 40 cm long and a diameter of 2.5 mm at a height of 70 cm above the floor. Under the center of the wire, a large box with sawdust was placed. For testing, the mouse was hung to the wire with the two limbs, and hang time was recorded. Each animal had three attempts for 180 sec maximum each. At the end of the test, the maximal holding impulse was calculated (hanging time × bodyweight).

At the study end (day 24), animals were sacrificed, and soleus (SO), tibialis anterior (TA), and extensor digitorum longus (EDL) muscle tissues were collected for subsequent molecular analysis. All experiments and procedures were approved by the local Animal Welfare Committee at Bienta and the Ukrainian Ministry of Education and Science (Ukrainian Ministry of Education of Science issued on 28-02-2019, registration number #1/11-2041).

### 2.3. NMR and 1RM Studies of Healthy Mice

Analysis of body composition of male C57BL/6N mice by nuclear magnetic resonance was performed as described [24]. Maximal muscle strength (1RM test, brief for Repetition Maximum) was determined by introducing a needle electrode directly into the M. tibialis anterior motor point (region of the common peroneal nerve). Subsequently, the muscle was electrically stimulated (4 V, 250 Hz, 2 s on and 1 s off) to certify the quality of contractions (full dorsiflexion). Based on previous observations, we loaded the muscles with 9% of total bodyweight and then stimulated with 8 V, 250 Hz, 2 s on and 1 s. Maximal loading was identified by observing one complete contraction without signs of fatigue or an incomplete range of motion, followed by an incomplete contraction. All experiments and procedures were approved by the local Animal Welfare Committee at the Institute of Biomedical Sciences, University of Sao Paulo (registration number #4356110320).

### 2.4. Protein Expression Analysis

Mass spectrometry (MS) based proteomic analysis was essentially performed as described earlier [22,23] using five EDL muscle tissues from compound-treated and non-treated Tu-muscles. MS raw data were processed as described by MaxQuant (1.6.6.0) using the Andromeda search engine and the Uniprot database for Mus musculus (as of 19 August 2019, containing 86,161 entries) [25]. At a false discovery rate of 1% [26] (both peptide and protein level; see also Appendix A), >2600 protein groups were identified. The reductive demethylation protocol employed [27] yielded pair-wise relative comparative ratios between proteins from Tu, Tu-946, Tu-205, and controls. Ratios were statistically queried for significant differences (see Supplementary Material Appendix A and Appendix A).

For Western blot analyses, frozen EDL was homogenized in Relax buffer (90 mmol/L N-2-hydroxyethylpiperazine-N-ethanesulfonic acid (HEPES), 126 mmol/L potassium chloride, 36 mmol/L sodium chloride, 1 mmol/L magnesium chloride, 50 mmol/L N-2-hydroxyethylpiperazine-N-ethanesulfonic acid (EGTA), 8 mmol/L adenosine triphosphate (ATP), 10 mmol/L creatine phosphate, pH 7.4) containing a protease inhibitor mix (Inhibitor mix M, Serva, Heidelberg, Germany) and sonicated. The protein concentration of the supernatant was determined (BCA assay, Pierce, Bonn, Germany), and aliquots (5–20 μg) were separated by Sodium Dodecyl Sulfate (SDS)-polyacrylamide gel electrophoresis. Proteins were transferred to a polyvinylidene fluoride membrane (PVDF) and incubated overnight at 4 °C using the following primary antibodies: anti-MuRF1 (Abcam ab183094, 1:1000), anti-MuRF2 (Myomedix, 1:1600), anti-ubiquitin (linkage-specific K48; Abcam ab140601, 1:1000), anti-LC3 B (Abcam 48394, 1:1000), anti-Nox2 (Abcam ab80508, 1:1000), anti-nitro-tyrosine (Abcam ab42789, 1:500). Membranes were subsequently incubated with a horseradish peroxidase-conjugated secondary antibody and specific bands visualized by enzymatic chemiluminescence and densitometry quantified using a software package, essentially as described previously [22,23]. Measurements were normalized to the loading control glyceraldehyde 3-phosphate dehydrogenase (GAPDH) (1/30000; HyTest Ltd., Turku, Finland). All data are presented as fold change relative to sham. 

### 2.5. Enzyme Activity Measurements

Skeletal muscle tissue was homogenized in Relax buffer, aliquoted, snap-frozen in liquid nitrogen, and stored at minus 80 °C for later enzyme activity measurements. Citrate synthase (CS, EC 4.1.3.7) and complex-I activities were determined, as previously described [28,29,30]. Enzyme activity data are presented as the fold change relative to control.

### 2.6. Statistical Analyses

Data are presented as mean ± SEM. One-way analysis of variance (ANOVA) followed by Bonferroni’s post hoc was used to compare groups, while two-way repeated-measures ANOVA followed by Bonferroni’s post hoc was used to assess bodyweight changes and contractile function (GraphPad Prism). Significance was accepted as *p* < 0.05. For proteomics data, an R pipeline (https://github.com/bhagwataditya/autonomics) based on limma, employed Bayesian-moderated *t*-testing on logarithmized and quantile normalized label-free quantification (LFQ) intensities to determine significant differences. Significance was accepted as *p* < 0.05 following multiple hypotheses testing corrections, according to Benjamini–Hochberg (see Supplementary Material Appendix A).

## 3. Results

### 3.1. B16F10-Melanoma Tumor Outgrowth Leads to Severe CaCax in Mice 

At days 9 and 10, mice were manually examined for tumor growth by palpation. No tumor outgrowth could be detected in five mice, i.e., in one from the Tu, two from the Tu-946, and in two mice from the Tu-205 group, respectively. These five mice were excluded from further analysis. The remaining 25 tumor-harboring mice were assessed by caliper for continued tumor growth at days 9, 10, 12, 15, 17, 19, 22. During this time, tumor volume increased in all three groups (Figure 1B). At day 21, after cell inoculation, a significantly lower tumor volume was seen in the Tu-946 group when compared to the untreated tumor group (Figure 1B). With respect to bodyweights, Tu mice had lost 3% at day 9, and 5% at day 12 of their bodyweights, respectively. This reduction was significant compared to con (p < 0.05; Figure 1C). WHTs, as a measure for muscle strength, performed at days 9, 16, and 23 showed an aggravating loss of holding impulse in Tu mice (Figure 2A). Therefore, we analyzed SO, EDL, and TA muscle weights on day 24 for further analysis of the underlying causes of muscle force loss in Tu. SO, EDL, and TA weights were significantly reduced, indicative of tumor-induced atrophy (Figure 2B–D). Detailed analysis at day 24 also indicated cardiac, kidney, thymus atrophy, epididymal fat loss, and splenomegaly (Figure 3A–E). Differential blood counts indicated severe anemia (Figure 3F). We conclude that the iliac B16F10 melanoma cell injection followed by 24 days tumor outgrowth in this murine model phenocopied muscle loss and weakness, as well as fat loss as found in human CaCax. These results are in good agreement with a recently published study by Voltarelli and colleagues [21].

### 3.2. Compound Feeding Improves Bodyweight, WHT Holding Impulse, and Epididymal Fat Content in the b16f10 Melanoma Model

Feeding with Myomed-946 and Myomed-205 attenuated Tu-induced weight loss: No significant bodyweight loss was apparent at day 9 in the Tu-946 and Tu-205 groups, respectively. At day 12, weight loss was significantly lowered (2%) compared to the 5% weight loss in the Tu-normal diet group (Figure 1C). Both compounds also protected holding impulse efficiently during tumor progression in WHTs at the measured time points on days 9, 16, 23 (Figure 2A). Attenuation of muscle strength by compound feeding goes along with preservation of tumor-induced loss of skeletal muscle mass (Figure 2B–D), especially in the EDL-205 group, where we noted the greatest effect on muscle mass preservation (Figure 2D). Finally, the addition of Myomed-946 and Myomed-205 to the mouse chow attenuated epididymal fat tissue loss (Figure 3D), protected from thymus atrophy (Figure 3C), and reduced splenomegaly (Figure 3E). However, both compounds had no impact on cardiac and kidney atrophy (Figure 3A,B), or anemia (Figure 3F). In summary, mice benefited from receiving compound diets during tumor outgrowth with regards to muscle strength, maintenance of bodyweight, skeletal muscle weight, and epididymal fat.

### 3.3. murf1 Inhibitor Treatment Attenuates CaCax Stress Signatures in the EDL Proteome

Next, we compared EDL control and Tu-proteomes by quantitative proteomic analyses (Appendix A). We selected EDL for proteomics because EDL weights are protected during Myomed-946 and -205 feeding (Figure 2D). A volcano plot comparison of EDL from healthy controls and BDF10 mice suggested specific alterations in EDL under melanoma-stress (see Appendix A), including Asph (a marker for tumor progression, maintenance, and also metastasis) [31,32], Pus7 (suggesting downregulation of protein synthesis [33]), Myl1 (myosin light chain 1; as recently also observed in human colon cancers and suggested as a marker for skeletal muscle cancer affection [34]), and Erc1 and Smdt1 (see Appendix A), both suggesting perturbed calcium-regulated cellular transport pathways: Erc1 belongs to the E, L, K, S protein rich (ELKS) family involved in Ca-dependent recruitment of vesicles to the synaptic cleft for exocytosis [35]; Smdt1 is a regulatory subunit of the mitochondrial uniporter that imports calcium into the mitochondrion [36]. Augmented Smdt1 may direct the mitochondrial Bax-pathway towards apoptosis [37]. Endoplasmic reticulum (ER) stress in Tu-EDL was suggested by Jagn1 and Sec62 (Appendix A): Jagn1 is induced by ER stress and connects to insulin signaling [38]; Sec62 connects the unfolded protein response (UPR) pathway to autophagosome activity [39]. Overall, the proteome signatures were consistent with suppressed translational machinery, metabolic and ER stress in Tu-EDL.

Compound feeding attenuated stress signatures (Appendix A): Myl1 and Erc1 were partially normalized. Components of the inner mitochondrial membrane responded to compounds, including Cox3, Cpt2, Qil1, and Lyrm4 [40,41,42,43]). Jagn1 and Sec62 were lower after compound feeding, suggesting lowered ER stress [38,39]. 

### 3.4. Compound Feeding Protects Complex-1 and CS Mitochondrial Enzyme Activities in TU-EDL

Compounds may affect energy metabolism according to our earlier data in cardiac cachexia [22,23]. We, therefore, next assessed the enzymatic activity of mitochondrial citrate synthase (CS) as a rate-limiting enzyme in the trichloroacetic acid (TCA) cycle that locates downstream of Cpt2 with regards to Acetyl-CoA metabolism and of respiratory mitochondrial complex I. Complex 1 is like Smdt1, Qil1, and Cpt2 (see Supplement Appendix A), a component of the inner mitochondrial member where it connects oxidative phosphorylation to mitochondrial membrane potential. Again, we used EDL extracts for CS and complex assays because EDL is protected during tumor outgrowth by both compounds (Figure 2D). Extracts from Tu-EDL had depleted enzyme activities for both CS and complex I (Figure 4). Compound feeding improved or rescued from this.

### 3.5. Compound Feeding to TU Mice Decreases murf1, UPS-Associated Ubi-k48, and ROS Stress Markers

Next, we analyzed the pathways implicated by MS and earlier studies [22,23] by Western blots. MuRF1-specific antibodies indicated elevated MuRF1 in Tu mice (Figure 5A). No change was observed for MuRF2 expression (Figure 5B) when feeding with Myomed-205 normalized MuRF1 levels to those of controls, whereas no change was observed with Myomed-946 (Figure 5A). No impact of both substances was seen on MuRF2 expression (Figure 5B). Consistent with an activation of the UPS system by MuRF1, we detected elevated Ubi-K48 levels as a marker for UPS activity [44] in Tu mice (Figure 5C). Feeding of Myomed-205 to Tu-mice normalized Ubi-K48 to control levels (Figure 5C). With regards to Nox2 and nitro-tyrosine, markers for reactive oxygen species (ROS) [45], antibodies detected elevated levels in Tu-EDL, and partial normalization during feeding (Figure 5D,E). Finally, LC3 I/II, a marker for autophagy [46], showed the reverse pattern, i.e., depletion in Tu mice, and normalization to control levels after compound feeding (Figure 5F). In conclusion, our Western blot studies suggested that Tu-EDL is subject to UPS, autophagosome- and ROS stress, and compound feeding attenuated stress responses.

### 3.6. Twenty-Four-Day Feeding with Myomed-205 Increases Lean Mass, Body Fat Content, and Muscle Strength in Normal Mice

Based upon the protection of epididymal fat (Figure 3D) and effects on Cpt2 (Appendix A), a gene involved in fat oxidation [41], we hypothesized that the compounds may affect fat metabolism and body composition in normal mice without stress. For testing this, we assessed the effects of a 1 g/kg Myomed-205 diet given for 24 days on whole-body fat content (we selected MyoMed-205 for this because its effects were more prominent). Food intake was not affected when compared to control (Figure 6A). However, mice receiving Myomed-205 food gradually gained more weight (~6% bodyweight at day 23, see Figure 6B). Analysis of animal at day 23 by NMR imaging indicated that this weight increase was attributed to an increased fat content (~17%, Figure 6C). Accordingly, lean mass was also increased by after 23 days of Myomed-205 (~6%, Figure 6D).

Based upon the WHT holding impulse strengthening in Tu mice (Figure 2A), we also tested if Myomed-205 might affect muscle force in healthy mice, possibly by effects on S100A1 (Appendix A), or Erc1, or Camk2d (Appendix A), genes involved in Excitation Contraction Coupling (ECC) coupling. To test this, we determined TA muscle force in the so-called 1RM test: The 1RM assay determines the force amount that a muscle can maximally develop to resist external stretch in its first bout before fatigue mechanisms may come into effect. Strikingly, the maximal 1R force in TAs from the Myomed-205 fed group was elevated by about 50% (Figure 6E).

## 4. Discussion

We previously studied the first prototype MuRF1 inhibitor in murine models for cardiac cachexia. There, chronic heart failure (CHF) was induced either by monocrotaline leading to right ventricular failure [22] or by myocardial infarcts leading to left ventricular failure [23]. In both models, CHF was accompanied by progressive wasting of peripheral skeletal muscles over 12-weeks. Treatment over the 12-week study period slowed down TA atrophy most prominently, and importantly, preserved diaphragm myofiber strengths, thereby attenuating these features of cardiac myopathy [22,23]. Here, we studied in a cancer cachexia model of two further optimized compounds from this novel drug class ([2-oxo-chromen-7-yl)heteromethyl]benzoic acids) because cancer cachexia represents an unmet medical need where no effective treatment is available to date. As an animal model, we selected BD16F10 melanoma-induced cachexia because tumor outgrowth has a marked effect on muscle strength in this murine model, as detected in WHTs from day 9 onwards (Figure 2A), and MuRF1 was upregulated in Tu-stressed EDL skeletal muscles (Figure 6A). As compounds, we selected Myomed-205 and -946, two currently prioritized compounds based on their toxicology data and improved serum stability (details on MyoMed-205 and -946 and their pharmacokinetics will be presented elsewhere). 

### 4.1. Compound Effects on Tumor Growth

The study groups that received B16F10 cell injections were initially not treated with compounds during days 1–3: We were concerned that compounds might possibly interfere with tumor formation, thereby complicating the interpretation of any results during the outgrowth phase. We hypothesized that the critical steps during tumor seeding were likely to be completed by around day 3, because 48 h later, small tumors can already be detected by palpation. Indeed, tumor seeding was high and did not differ statistically between the compound or normal diet-fed groups, i.e., 9/10, 8/10, and 8/10 mice developed tumors in groups, respectively. Next, we determined if compounds had an effect on tumor outgrowth. Myomed-946 appeared to slow down tumor outgrowth somewhat (*p* < 0.05 at day 24, 30% mass reduction). Comparison of the MyoMed-946 and MyoMed-205 proteome data sets suggests that MyoMed-946-treated EDL may contain higher levels of Ttll12 (tubulin-tyrosine ligase-like protein 12) and Tbc1d17, both regulators of tubulin dynamics [47,48], correlating in ovarian cancers with a poor outcome [49]. Possibly, MyoMed-946, in contrast to MyoMed-205, rapidly affects dividing tumor cells via Ttll12 and Tbc1d17, a hypothesis that has to be validated in future studies. Myomed-205 feeding had no effect on the B16F10 tumor growth curve (Figure 1B). Therefore, we decided to focus on Myomed-205 when addressing the compound’s effects in healthy mice.

### 4.2. Compound’s Effects on Cacax and EDL

Both Myomed-946 and Myomed-205 had beneficial effects on Tu-mice with regard to cachexia and myopathy: Compound feeding protected from tumor-induced bodyweight loss at days 9 and 12 (Figure 1C). During tumor outgrowth, compound feeding slowed SO, TA, and EDL muscle weight loss until day 24 (Figure 2B–D). Holding impulse strengths as monitored by WHTs on days 9, 16, and 23 were efficiently protected (Figure 2A).

Dissections at day 24 indicated compound effects extend beyond muscle: Both compounds protected from epididymal fat loss, attenuated thymus atrophy, and improved splenomegaly (Figure 3A–E). These non-muscle effects were unexpected as we hypothesized at the project start that compound effects are from interference with upregulated MuRF1 in Tu and, thus, are specific for stressed skeletal muscles. To follow up on non-muscle effects, we decided to also feed healthy and non-stressed mice with Myomed-205. This increased whole bodyweight, lean mass, and fat content over 24 days, while food intake remained normal (Figure 6A–D). Muscle strength, as determined by the 1RM test, markedly increased as well (Figure 6E). Therefore, the feeding of normal mice overall recapitulates major effects observed in Tu mice, i.e., an anabolic compound effect on bodyweight, a stimulation of lipogenesis, and augmented muscle force (i.e., augmented 1RM force). Therefore, Myomed-205 could be a pro-anabolic agent, and studies of some anabolic pathway constituents should be done in a future study to confirm or refute this hypothesis.

Future studies will need to compare the effects of MyoMed-205 and MyoMed-946 on muscle weight directly, such as if mass is better preserved during MyoMed-205 treatment, perhaps in line with a better efficacy on mitochondrial activities. However, the present study was designed for a statistically significant comparison between TU and treatment groups, but not between the two treatment groups for relative differences between compounds. Future studies will need to address this issue with higher animal numbers if a better improvement in mitochondrial complex I activity (Figure 4B) and a greater reduction of MuRF1 expression (Figure 5A) also relate to improved muscle weight and/or functions by MyoMed-205.

As an initial attempt to characterize the cellular mechanisms that underlie compound effects, we have carried out proteomic profiling, enzymatic assays, and western blot studies. Overall, our results suggest that Tu- stress in EDL adversely affects protein translation, mitochondrial energy metabolism, and UPS-driven catabolism. Myomed-205 or -946 protect from this, including the rescue of mitochondrial complex-1 and CS activities (Figure 4), and attenuation of the MuRF1-UPS axis (Figure 5). Future studies will need to address the relevance of effects on Smdt1 (a protein that activates the Bax pathway via enhancing mitochondrial uniporter mediated calcium uptake [37]), the ER stress marker Sec62, and the pro-tumorigenic enzyme Asph [32,38,50] (Appendix A).

## 5. Conclusions and Future Perspectives

Recent findings in cancer cachexia research imply skeletal muscles [1], and, in particular, impairment of their mitochondria [2] as the key to the progression of wasting during cancer growth. Accordingly, the main relevance of the present study is that Myomed-205 can preserve metabolic functions in stressed myocytic mitochondria during tumor outgrowth. Another possible medical implication of our study is that since Myomed-205 and Myomed-946 have no relationship to other known drugs, their combination with other treatment strategies, such as anabolic growth hormones [3] or neurohumoral stimulants [4], appears feasible. Next, it remains to be determined which mechanisms account for the marked strength increase and metabolic effects in Tu and healthy mice fed with Myomed-205 or -946. With regard to muscle strength, Myomed-205 possibly acts via S100A1, and Erc1, a SR-calcium release regulating factor that accounts for muscle weakness in Lambert–Eaton syndrome [51]. With regard to fat metabolism, transgenic MuRF1 mouse studies have implicated MuRF1 in the regulation of fat metabolism via pyruvate dehydrogenases (PDH) and pyruvate dehydrogenase kinase (PDK)-4 [52]. Since the used [2-oxo-chromen-7-yl) heteromethyl]benzoic acids are not related to other known drug classes, it also remains to be determined if co-treatment strategies may be warranted, such as with MT-102 that was recently reported to provide benefits to cancer cachexia by targeting the adrenergic system [53]. Future studies have to identify the exact molecular mechanisms of how MyoMed-205 or MyoMed-946 provoke the beneficial effect seen in several muscle-wasting conditions, such as pulmonary hypertension [22], chronic heart failure [23], and cancer cachexia. 

## 6. Patents

V.A. and S.L. have a patent application pending for [2-oxo-chromen-7-yl)heteromethyl]benzoic acids and their application to chronic muscle stress states.

## Figures and Tables

**Figure 1 cells-09-02272-f001:**
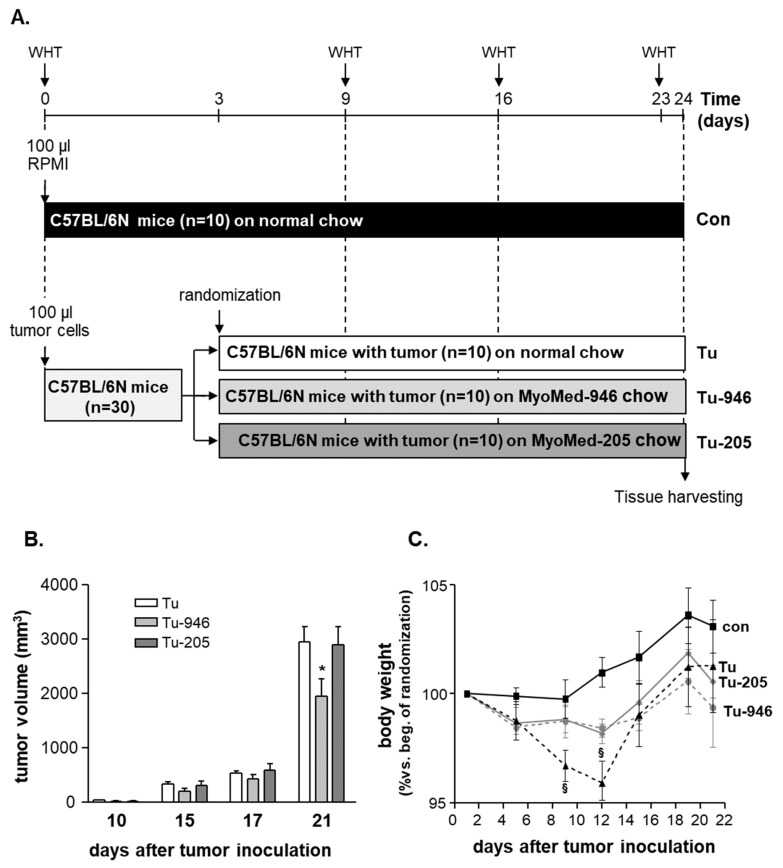
Study design for inducing melanoma tumor growth followed by monitoring cancer cachexia (**A**). Thirty mice were injected with 5x105 melanoma cells (B16F10 cell line). As controls, ten mice were injected with RPMI medium only without tumor cells (con). The B16F10 inoculated mice were randomly assigned on day 3 to study groups either receiving standard diet during melanoma outgrowth (Tu), or received food supplemented with Myomed-946 (Tu-946), or receiving chow supplemented with Myomed-205 (Tu-205), respectively. Wire hang tests (WHTs) were performed at days 9, 16, and 23. Mice were sacrificed at day 24 for dissections. Tumor growth was monitored non-invasively by caliper measurements (**B**) and animal weights (**C**) were assessed. Values are shown as mean ± standard error of the mean, * *p* < 0.05 vs. Tu at the same time point, ^§^
*p* < 0.05 vs. con at the same time point.

**Figure 2 cells-09-02272-f002:**
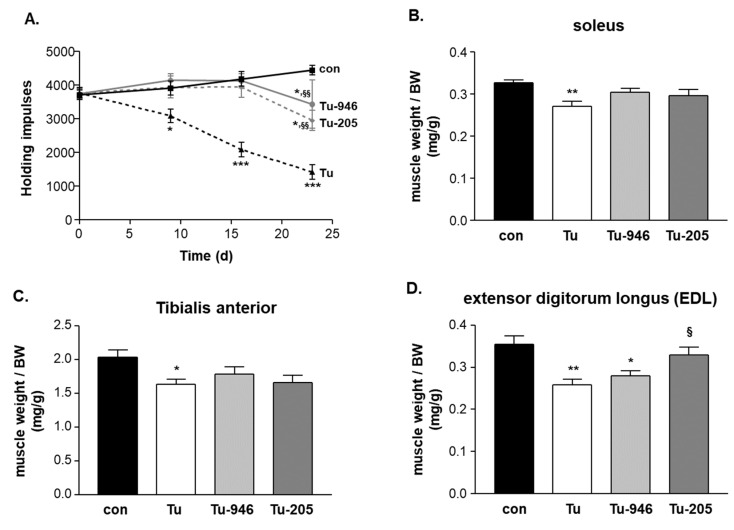
Effect of Myomed-946 and -205 on skeletal muscle weights and wire hang test (WHT) expressed as maximal holding impulses in Tu mice. WHTs indicated progressively falling holding impulses until day 23 in the Tu group. Feeding with Myomed-946 and -205 attenuated this (**A**). Soleus (**B**) tibialis anterior (**C**) and EDL (**D**) muscle weights were determined at day 24 when dissecting the four study groups. Tumor outgrowth lowered muscle weights in all three skeletal muscles types. Feeding with Myomed-946 and -205 attenuated muscle wasting. Values are shown as mean ± standard error of the mean, * *p* < 0.05, ** *p* < 0.01 *** *p* < 0.001 vs. con; ^§^
*p* < 0.05, ^§§^
*p* < 0.01 vs. Tu.

**Figure 3 cells-09-02272-f003:**
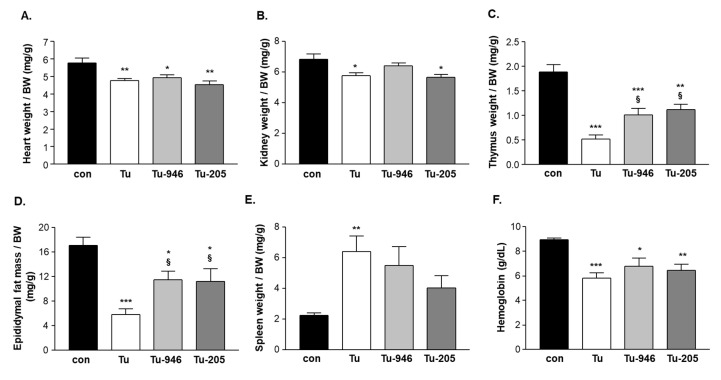
Effects of tumor outgrowth and feeding of Myomed-946 and -205 on heart (**A**), kidney (**B**), thymus (**C**), epididymal fat (**D**), and spleen weight (**E**) by day 24. Tumor outgrowth caused severe loss of epididymal fat, thymus atrophy, and splenomegaly. These effects were significantly offset by Myomed-946 or -205 feeding. Kidney and cardiac atrophy during tumor outgrowth were not significantly prevented by the compounds and remained on Tu levels below con. Clinical chemistry detected severe anemia in Tu mice as apparent by low hemoglobin (Hb) content (**F**). Values are shown as mean ± standard error of the mean, * *p* < 0.05, ** *p* < 0.01, *** *p* < 0.001 vs. con; ^§^
*p* < 0.05 vs. Tu.

**Figure 4 cells-09-02272-f004:**
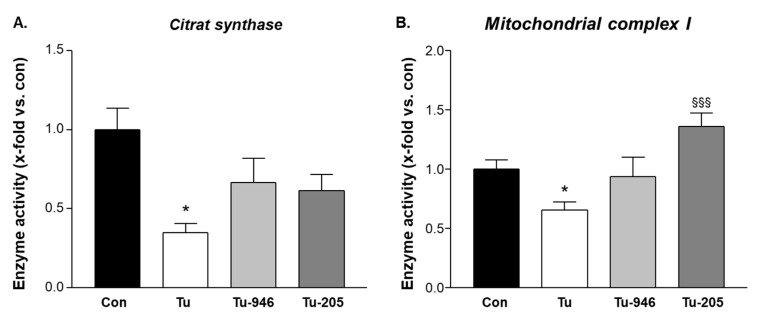
Citrate synthase and complex I activity in groups #1-4 (**A**). Melanoma outgrowth leads to depleted CS and complex 1 activity. Feeding with Myomed-946 (TU-946) or Myomed-205 (TU-205) reverses this reduction (**B**). Values are shown as mean ± standard error of the mean, * *p* < 0.05 vs. con; *^§§§^ p* < 0.001 vs. Tu.

**Figure 5 cells-09-02272-f005:**
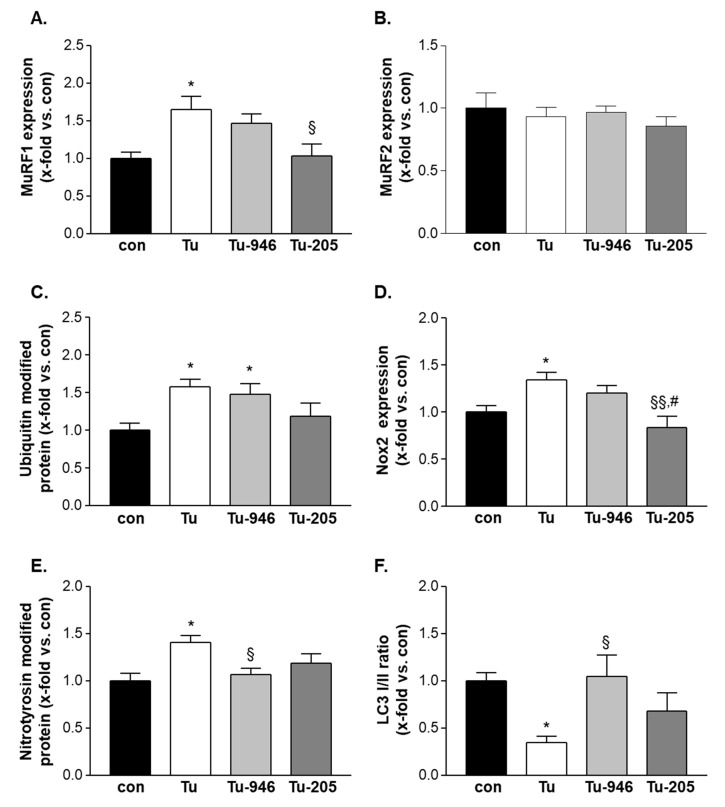
Western blot analysis were performed in the EDL muscle in all groups using specific antibodies against MuRF1 (**A**), MuRF2 (**B**), ubiquitin modified proteins (**C**), Nox2 (**D**), nitrotyrosin modified proteins (E) and LC3 (F). Values are shown as mean ± standard error of the mean. * *p* < 0.05 vs. con; ^§^
*p* < 0.05, ^§§^
*p* < 0.01 vs. Tu; # *p* < 0.05 vs Tu-946.

**Figure 6 cells-09-02272-f006:**
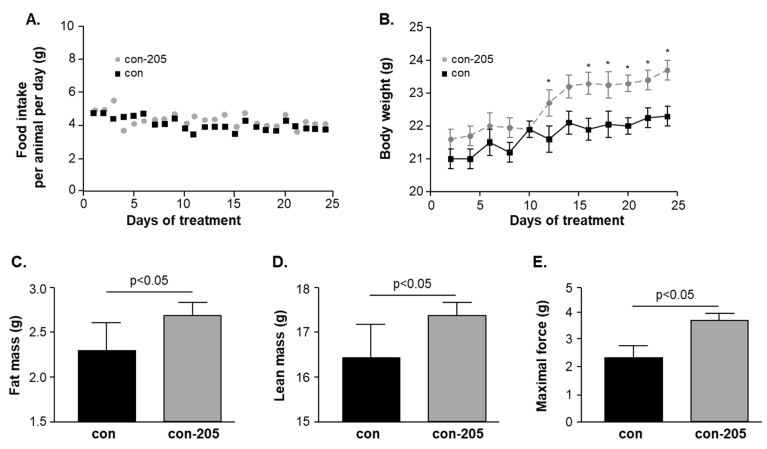
Effects of 24-day Myomed-205 feeding on healthy control mice. Supplementation of food with 1g Myomed-205 per kg standard diet did not affect food intake over 24 days (**A**). The Myomed-205 fed mice (con-205) had further elevated body weight (~6%) at day 24 when compared to control mice (con). Note that significant differences emerged at day 10 (**B**). Myomed-205 feeding increased total body fat by approximately 17% (**C**), total lean mass by 6% (**D**) and maximum muscle force by 50% (**E**) at day 24. Values are shown as mean ± standard error of the mean, * *p* < 0.05 vs. con.

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
