# Peer review of "Small-Molecule Chemical Knockdown of MuRF1 in Melanoma Bearing Mice Attenuates Tumor Cachexia Associated Myopathy"

_cells, 2020, doi:10.3390/cells9102272_

Round 1

Reviewer 2 Report

Summary

This manuscript reported the effects of MuRF1 inhibitors, Myomed-205 and Myomed-946, on the melanoma-induced cancer cachexia (CaCax) model mice. The orally administered these inhibitors protects the deteriorations of muscle volume, strength, and mitochondrial energy metabolism with downregulating MuRF1 pathway. The authors concluded that these MuRF1 inhibitors can be a beneficial treatment for CaCax. Almost experiments seem to be performed and presented well but some minor mistakes pointed below need to be corrected. And we encourage the authors to discuss the differences between Myomed-205 and -946 in detail for clinical application.

Comments

  1. P3, lines 28-29: The Introduction does not contain the words “Myomed-205 and Myomed-946”. I think the molecular names are better to be described in the Introduction.
  2. Methods and Materials: The information of the standard rodent chow, Myomed#946, and Myomed#205 used in this study need to be described (manufacture or something).
  3. Comprehensively, the effects of Myomed-205 were more beneficial rather than those of Myomed-946. Please discuss the reasons of these differences based on molecular structures or processes. It will provide useful information to apply Myomed-205 and -946 for clinical research.

Minor points

  1. P4, lines 4 and 5: “100μl” should be “100 μl” (insert a space).
  2. P4, line 4: “fetal calf serum” should be “FCS”.
  3. “C57/BL6N” in P4 line 2, “C57BL/6N” in P5 line 3, “C57Bl/6N” in Figure 1. Which is correct?
  4. Figures 1A, 4A, and 4B: What is “MCEMBL-205”?
  5. Figures 1B and 2A: What is “Tu-704946”?
  6. P6, line 5: “Sao Paulo and the”?
  7. P6, line 20: “20μg” should be “20 μg” (insert a space).
  8. P6, line 28: “previously.18,19” should be “previously [18,19].”.
  9. P7, lines 4-18: Italic text need to be corrected.
  10. Figures 4C, 4D, and 4E: What is “con-205”?

Reviewer 3 Report

The work presented by Volker Adams and colleagues studied skeletal muscle atrophy and dysfunction in the murine cancer cachexia (CaCax) model by injecting B16F10 melanoma cells into mouse thighs and following mice during melanoma outgrowth. They also found that inhibition of MuRF1 (by Myomed-205 and -946) attenuate the above effects. This study is novel, and the manuscript is well written. However, I just have some minor concerns.

  1. Please expand the introduction section a little more on the following points: which cancer types are more prone to cancer cachexia related death? the rationale for choosing this particular model for your study?
  2. What could be the reason for the reduction of tumor volume by Myomed-946, but not for the Myomed-205 treatment, please explain that in the discussion section.
  3. Please provide the images of the western blots for Figure 6, and what internal controls were used?
  4. Please expand the discussion section with respect to recent findings on cancer cachexia?
  5. Please write a conclusion from the overall results of this study after the discussion section. And also write a little bit about future directions.

Round 2

Reviewer 1 Report

The manuscript by Adams et al “Small-molecule chemical knock-down of MuRF1 in 2 melanoma bearing mice attenuates tumor cachexia 3 associated myopathy”, reports the potential beneficial effect of two small molecules, Myomed-205 and Myomed-946, on tumor-induced cachexia. These molecules attenuated the induction of MuRF1 in tumor stressed muscle, increased muscle performance and attenuated skeletal muscle loss. They also protected complex-1 and Citrate Synthate mitochondrial enzyme activities. Overall this paper is interesting and provides new information, but some major points must be addressed by the authors.

There are several inaccuracies throughout the text that need to be clarified:

1/ The title should be modified since this molecule has been shown to down-regulate MuRF1 and MuRF2 expression (Adams 2019 JCSM): “Small-molecule chemical knock-down of MuRF and MuRF2 in melanoma bearing mice attenuates tumor cachexia associated myopathy”.

2/ in the abstract lane 10: “Myomed-205 and -946 that inhibit the E3 ligase MuRF1.” The authors should precise “inhibit MuRF1 activity and MuRF1 and MuRF2 expression”

3/ In the introduction part, some references are missing

        - lane27, “… this in turn attenuates the loss of contractile proteins.”

The cited references concern only the proteins of the thick filament and not of the thin filament. However, muscle atrophy induced by MuRF1 is due to the degradation of both thick and thin filaments. Therefore, the authors should at least add the following references: Kedar et al. PNAS 2004 (doi: 10.1073/pnas.0404341102) and Polge et al. FASEB J. 2011 (doi: 10.1096/fj.11-180968).

4/ The authors should give more details about the published characteristics of their molecules, as the shortcuts used could lead to misinterpretation:

     4.1 Lane 28. As the authors used recently identified and published molecules, they must specify the "former" name of the molecules, the one used in the previous publications (Bowen et al JCSM 2017, and Adams et al 2019). So that readers can easily find the effects observed for these molecules in these previous publications. Indeed Tu-296 is probably derived from the small molecule 704296 (Bowen et al JCSM 2017), this should be precise. However, it is less clear for Tu-205.

      4.2 lane 29.  “We hypothesized that recently discovered small molecules from a titin-MuRF1 interaction high-throughput screen that attenuated skeletal muscle atrophy in murine cardiac cachexia models”. This sentence should be rephrased since it suggests that only the lack of interaction between MuRF1 and titin leads to the phenotype observed. The authors should precise that these small molecules have an inhibitory effect on the expression levels of MuRF1 and MuRF2 and that this effect take probably an important part in the observed phenotype and could alone explain it (Bowen et al JCSM 2017, and Adams et al 2019).

      4.3 lane 29. In the same sentence “…attenuated skeletal muscle atrophy in murine cardiac cachexia models ”. This sentence suggest that all the skeletal muscles were spared in these models, whereas no protective effect was observed on EDL and soleus muscles, a mild protective effect was reported on Tibialis anterior mass, the most important effect being on the contractile function of diaphragm (Bowen et al JCSM 2017, and Adams et al 2019). The authors should add this additional information (both in the introduction and in the discussion (p12, Lane6) parts) to avoid misinterpretation by readers. Depending on the models studied, these molecules may not be equally effective in protecting all skeletal muscles.

5/ in the results part, the Cacax induced by B16F10 has already been described by Voltarelli et al (“Syngeneic B16F10 Melanoma Causes Cachexia and Impaired Skeletal Muscle Strength and Locomotor Activity in Mice”, Front. Physiol. 8:715. doi:10.3389/fphys.2017.00715). The results presented in paragraph 3.1 should therefore be presented as a validation of the model (with reference to Voltarelli) rather than as original results.

Regarding the model:

- The authors should explain why they choose this model (among other) to induce Cacax and give some references.

- The authors should precise if the injections of melanoma cells were intra-muscular or subcutaneous.

- Since the B16F10 cells are highly metastatic, did the authors check for liver or spleen metastasis?

In the results part:

- The authors should explain how they chose the dose of inhibitor studied (0.1% of normal chow). Have there been any pre-tests performed, with several doses? Was the DL50 measured on cells? This information is important and has not been provided in this or previous publications.

- The mRNA expression level of MuRF1 and MuRF2 in the different groups should be presented since this molecule affect the expression of both.

Minor points

In the results part:

- P9, lane14. The authors should move the paragraph 3.4 to the end of the results since it concerns the effect of one of the molecules on normal mice, while the analysis on cacax mice continues after this paragraph. This could be confusing.

- P10, lane 4. “effect” should be changed by “affect”

In the discussion part

- p12, lane 7 : See comment 4.3

- The author should add that Myomed-205 could be a pro-anabolic agent regarding the results they observed in the paragraph 3.4 on normal mice. The study of some anabolic pathway constituents should be done in a future study to confirm or refute this hypothesis.

Figures:

- The figures 2, 3 and 4 and their legends are difficult to read, they should be improved. For example, in the fig2A, what is written to the right of the curves is totally illegible and blurred.

- In Figures 5 and 6, it is not clear what the significant results are, which may cause confusion or lead to false conclusions/impressions. This is especially the case for the results obtained with Tu-205. Authors should add symbols above the histogram bars with significant differences (as was done previously e.g. fig3).

- Different names are used for the two small molecules studied in Figures 1, Fig2A, fig4A, 4B.
